Evaluation of the glycemic effect of Ceratonia siliqua pods (Carob) on a streptozotocin-nicotinamide induced diabetic rat model

Qasem Mousa A. mousa505281@gmail.com 1
Noordin Mohamed Ibrahim ibrahimn@um.edu.my 1
Arya Aditya 2
Alsalahi Abdulsamad 3
Jayash Soher Nagi 4 5
1 Department of Pharmacy, Faculty of Medicine, University of Malaya , Kuala Lumpur , Malaysia
2 Department of Pharmacology and Therapeutics, School of Medicine, Faculty of Health and Medical Sciences, Taylor’s University , Subang Jaya , Malaysia
3 Department of Pharmacology, Faculty of Medicine, University of Malaya , Kuala Lumpur , Malaysia
4 Department of Restorative Dentistry, Faculty of Dentistry, University of Malaya , Kuala Lumpur , Malaysia
5 Department of Oral Medicine and Periodontology, Faculty of Dentistry, Ibb University , Ibb , Yemen
Nogueira Cristina
Electronic publication date: 2018 May 23
Publication date: 2018
Volume: 6
Electronic Location ID: e4788
Received 2018 Mar 7; Accepted 2018 Apr 27
Copyright: ©2018 Qasem et al.
Copyright year: 2018
Copyright holder: Qasem et al.
License: This is an open access article distributed under the terms of the Creative Commons Attribution License, which permits unrestricted use, distribution, reproduction and adaptation in any medium and for any purpose provided that it is properly attributed. For attribution, the original author(s), title, publication source (PeerJ) and either DOI or URL of the article must be cited.
License URL: https://creativecommons.org/licenses/by/4.0/

Keywords: Carob, Ceratonia siliqua, Type 2 Diabetes Mellitus, α-amylase, α-glucosidase, Acute oral toxicity, Cytotoxicity, HOMA-IR

Funding: University of Malaya Research Grant RP001D-13BIO This research was made possible by the support of a University of Malaya Research Grant (UMRG Grant: RP001D-13BIO). There was no additional external funding received for this study. The funders had no role in study design, data collection and analysis, decision to publish, or preparation of the manuscript.

==============================
Background

Ceratonia siliqua pods (carob) have been nominated to control the high blood glucose of diabetics. In Yemen, however, its antihyperglycemic activity has not been yet assessed. Thus, this study evaluated the in vitro inhibitory effect of the methanolic extract of carob pods against α-amylase and α-glucosidase and the in vivo glycemic effect of such extract in streptozotocin-nicotinamide induced diabetic rats.

Methods

2,2-diphenyl-1-picrylhydrazyl (DPPH) and Ferric reducing antioxidant power assay (FRAP) were applied to evaluate the antioxidant activity of carob. In vitro cytotoxicity of carob was conducted on human hepatocytes (WRL68) and rat pancreatic β-cells (RIN-5F). Acute oral toxicity of carob was conducted on a total of 18 male and 18 female Sprague-Dawley (SD) rats, which were subdivided into three groups (n = 6), namely: high and low dose carob-treated (CS5000 and CS2000, respectively) as well as the normal control (NC) receiving a single oral dose of 5,000 mg kg−1 carob, 2,000 mg kg−1 carob and 5 mL kg−1 distilled water for 14 days, respectively. Alkaline phosphatase, aspartate aminotransferase, alanine aminotransferase, total bilirubin, creatinine and urea were assessed. Livers and kidneys were harvested for histopathology. In vitro inhibitory effect against α-amylase and α-glucosidase was evaluated. In vivo glycemic activity was conducted on 24 male SD rats which were previously intraperitoneally injected with 55 mg kg−1 streptozotocin (STZ) followed by 210 mg kg−1nicotinamide to induce type 2 diabetes mellitus. An extra non-injected group (n = 6) was added as a normal control (NC). The injected-rats were divided into four groups (n = 6), namely: diabetic control (D0), 5 mg kg−1glibenclamide-treated diabetic (GD), 500 mg kg−1 carob-treated diabetic (CS500) and 1,000 mg kg−1 carob-treated diabetic (CS1000). All groups received a single oral daily dose of their treatment for 4 weeks. Body weight, fasting blood glucose (FBG), oral glucose tolerance test, biochemistry, insulin and hemostatic model assessment were assessed. Pancreases was harvested for histopathology.

Results

Carob demonstrated a FRAP value of 3191.67 ± 54.34 µmoL Fe++ and IC50 of DPPH of 11.23 ± 0.47 µg mL−1. In vitro, carob was non-toxic on hepatocytes and pancreatic β-cells. In acute oral toxicity, liver and kidney functions and their histological sections showed no abnormalities. Carob exerted an in vitro inhibitory effect against α-amylase and α-glucosidase with IC50 of 92.99 ± 0.22 and 97.13 ± 4.11 µg mL−1, respectively. In diabetic induced rats, FBG of CS1000 was significantly less than diabetic control. Histological pancreatic sections of CS1000 showed less destruction of β-cells than CS500 and diabetic control.

Conclusion

Carob pod did not cause acute systemic toxicity and showed in vitro antioxidant effects. On the other hand, inhibiting α-amylase and α-glucosidase was evident. Interestingly, a high dose of carob exhibits an in vivo antihyperglycemic activity and warrants further in-depth study to identify the potential carob extract composition.

Introduction

Diabetes mellitus (DM) is a widespread metabolic disorder that is estimated to affect 640 million people by 2040 (International Diabetes Federation, 2016). DM occurs owing to a defect in either insulin secretion, insulin action, or both leading to an increase in blood glucose (Gamboa-Gómez et al., 2017). Accordingly, DM is classified into either type 1 diabetes mellitus (T1DM) which is insulin dependent or type 2 diabetes mellitus (T2DM) which is non-insulin dependent (WHO, 2016). However, T2DM is the most common type of DM constituting 90–95% of all cases of DM worldwide (Ahmed et al., 2016; Tripathi, 2013).

T2DM is mainly managed with oral hypoglycemic agents. However, due to the potential side effects of those agents, World Health Organization (WHO) recommends using alternative medicines (e.g., natural products) (Abdel-Sattar et al., 2013) as a second choice (Saghir et al., 2016). As a matter of fact, more than 1,200 species of medicinal plants could be used as an antidiabetic (Abdel-Sattar et al., 2013; Alarcon-Aguilar et al., 2002; Hernandez-Galicia et al., 2002).

Carob (Ceratonia siliqua L) is an evergreen plant (Roseiro et al., 2013), which was reported as a native plant to Yemen (Battle & Tous, 1997). Traditionally, C. Siliqua (carob) is used as an antitussive, antidiarrheal and a diuretic (Baytop, 1984; Gulay et al., 2012; Merzouki et al., 1997). However, C. Siliqua pods (Carob) were reported to exert several pharmacological properties such as antioxidant, anti-ulcer and anti-inflammatory (Kivcak, Mert & Ozturk, 2002; Rtibi et al., 2015a; Rtibi et al., 2017a; Rtibi et al., 2016).

In folk Yemeni medicine, diabetics claim that carob pods can reduce their elevated blood sugar. However, no study has been conducted to explore the glycemic effect of the Yemeni cultivar of carob pods. Consequently, this study sought to evaluate the glycemic and the antioxidant activities of the methanolic extract of carob in streptozotocin-nicotinamide induced diabetic rat model.

Materials and Methods

Materials and chemicals

Streptozotocin (Merck Millipore, Temecula, CA, USA), nicotinamide, sodium citrate monohydrate, phosphate-buffered saline (PBS) and glucose anhydrous (Sigma-Aldrich, St. Louis, MO, USA). Citric acid (Merck, Kenilworth, NJ, USA), glibenclamide (Sigma-Aldrich, USA), formaldehyde (HmbG chemicals, Hamburg, Germany), one-touch glucometer (Accu-Chek Performa, Roche, Mannheim, Germany). Rat Insulin ELISA Kit (ER1113, Wuhan Fine Biological Technology Co., Ltd., China). 2,4,6-Tri (2- pyridyl)-s-triazine, Folin-Ciocalteu reagent and ferric chloride (Sigma-Aldrich, USA). Cellulose extraction thimble (Tokyo Roshi Kaisha, Ltd, Tokyo, Japan), ferrous sulphate heptahydrate (Essex, UK). Amylase and glucosidase enzymes, 3, 5-dinitrosalicylic acid (DNSA) and 4-Nitrophenyl (3-D-glucopyranoside (P-NPG) substrate, monobasic potassium phosphate and potassium tartrate tetrahydrate (Sigma-Aldrich, USA), α-acarbose 95% (Acros organic, USA). Potato starch (Sigma-Aldrich, USA).

Collection of plant material

Totally, two kilograms of unripe pods of carob were collected from Bani Yousef region, 70 km from Taiz governorate, Yemen. The pods were collected at 8 am, 15th May 2014. The shoots were wrapped in aluminum foil sheets and put in ice-filled box to isolate it from sunlight and external temperature. The unripe carob samples were transported directly to the Laboratory of Pharmacognosy Department, Faculty of Pharmacy, Sana’a University, Yemen. The plant was verified and authenticated by a taxonomist and given a voucher specimen No. (CSL/2014/8/1), which was deposited at the Laboratory of Pharmacognosy, Faculty of Pharmacy, Sana’a University, Yemen.

Extraction

The unripe carob pods were washed with purified water to eliminate debris and dried with a fan. Then, the pods were chopped to small pieces and left to dry at room temperature (28.0 ± 2 °C) for three weeks. Next, chopped pieces were ground with a grinder to get fine powder at department of Pharmacy, Faculty of Medicine, University of Malaya. One kilogram of carob powder was extracted using Soxhlet method as previously described with some modification (Jadhav et al., 2009). Extraction process was performed by methanol using Soxhlet extractor under 50 °C. Filter paper (Whatman No. 1) was used to filter the liquid extract. Rotavapor evaporator was used to separate the solutes from the solvents at 40°C and the filtrate was allowed to dry in dark place. Finally, the dried filtered extract was kept under −20 °C for later work.

In vivo and in vitro experimental model design of study

Fig. 1 Flow chart of the experimental design

Figure 1 In vivo and in vitro experimental model design of study.

The flow chart of the experimental design.

Quantification of total phenolic and flavonoid content

Total phenolic content (TPC) of the methanolic extract of carob was determined using Folin-Ciocalteu assay (Chan et al., 2009). Gallic acid was used as a reference. Totally, 20 µL of the extract and gallic acid (1 mg mL−1) were pipetted into 96-well microplate followed by addition of 50 µL of 10% Folin-Ciocalteu reagent to be incubated for 3.0 min. Then 100 µL of sodium carbonate (10%) was added followed by incubation for one hour. The absorbance was measured at 765 nm spectrophotometrically (Infinite M 200; Tecan, Männedorf, Switzerland). The phenolic content was represented as mg of gallic acid equivalent/g extract.

Total flavonoid content (TFC) was estimated using AlCl3 method (Nabavi et al., 2008) in which 10 µL of the extract and quercetin were transferred into 96-well plate, followed by addition of 60 µL of DMSO. Then, 10 µL of 10% (w/v) aluminium chloride, 10 µL of 1 M potassium acetate and 120 µL of deionized water were added to the mixture, which was incubated for 30 min at room temperature. Serial dilutions of quercetin (500–31.25 µg mL−1) were prepared to generate the calibration curve. The absorbance was measured at 415 nm spectrophotometrically (Infinite M 200; Tecan, Switzerland). The flavonoid content was presented as mg quercetin equivalent/g extract.

Evaluation of antioxidant properties of carob methanolic extract

Radical-scavenging activity of carob against free radicals of DPPH

Radical scavenging activity of the methanolic extract of carob was determined using the stable free radical 2,2-diphenyl-1-picrylhydrazyl (DPPH•) with slight modifications (Alshawsh et al., 2012). Briefly, 1.2 mg of DPPH was thawed out in 30 mL of DMSO to prepare 100 µM DPPH reagent solution. Stock sample (1.0 mg/1.0 mL DMSO) was diluted. Serial dilutions; 500, 250, 125, 62.50, 31.25, 15.62 and 7.18 µg mL−1 of stock sample were prepared. Totally, 25 µL of sample and 150 µL of DPPH solution were transferred into 96-wells plate. Next, the mixture was incubated in a dark place at room temperature for 25 min. The absorption was measured spectrophotometrically (Infinite M 200 Tecan, Switzerland) at 517 nm. Quercetin was used as a positive control. The amount of the extract was replaced by DMSO in case of negative control. Percentage inhibition was obtained using the following equation: %Inhibition=Ac−As∕Ac∗100.

where Ac is the absorbance of negative control, As is the absorbance of test sample.

The concentration of the methanolic carob extract that was able to inhibit 50% of the free radicals of DPPH (IC50) was obtained from the calibration curve of sample and positive control using a non-linear regression analysis and the results were expressed as an IC50 value ± SD. All determinations were conducted in triplicate to evaluate the IC50.

Ferric reducing antioxidant power assay (FRAP)

Measuring of ferric reducing antioxidant activity of the methanolic extract was performed according to (Benzie & Strain, 1996). Working FRAP reagent was prepared through mixing 300 mM acetate buffer (pH 3.6), 10 mM 2,4,6-tripyridyl-s-triazine (TPTZ) in 40 mM HCL, and 20 mM ferric chloride (FeCl3.6H2O). Calibration curve was plotted using FeSO4⋅7H2O serial concentrations between 100–1,000 mg mL−1. Freshly prepared FRAP working reagent was warmed at 37 °C for 5 min, then 30 µL of the extract, Trolox and quercetin (1.0 mg dissolved in 1.0 mL DMSO) were transferred into 96-well microplate and mixed with 300 µL of FRAP reagent. Then, absorbance was measured at 593 nm against the blank (DMSO). The values were represented as µmoles of ferrous sulphate equivalent/mg extract.

In vitro inhibitory effect of carob against α-amylase and α-glucosidase

The α-amylase inhibition capacity of the extract was determined according to Loizzo et al. (2007). In brief, 20 µL of each test sample (1 mg mL−1extract) dissolved in DMSO or standard Acarbose 95% (Acros Organic, Bridgewater, NJ, USA) and 50 µL 2U mL−1 of α-amylase enzyme (Porcine Pancreas Amylase; 2 mg/10 mL; 10 Units/mg; Sigma-Aldrich, USA) dissolved in cold deionized water were added into each well. After incubation for 10 min at 28 °C, 100 µL of starch solution (0.5% w/v (Sigma-Aldrich, USA) prepared by mixing 250 mg potato starch in previously prepared 50 mL phosphate buffer pH 6.9 prepared by mixing 20 mM monobasic sodium phosphate and 6.7 mM of sodium chloride and heated at 60 °C for 15 min) was added to the reacting mixture. Thereafter, the reaction mixture was re-incubated for 10 min at 28 °C and 100 µL of colorimetric DNSA reagent 96 mM (Prepared by mixing 12 g of sodium potassium tartrate tetrahydrate in 8 mL of 2 M NaOH and 0.5 mg 3,5-dinitrosalicylic acid solution) was added with mixing the contents well. The reaction was terminated by incubating the mixture in a water bath 86 °C for 15 min and later was cooled to room temperature. The blank was conducted in a similar method, with the replaced α-amylase enzyme by deionized water. All the reagents involved the α-amylase enzyme with the exception of the test extracts was pointed as a reference sample. The absorbance of reaction was measured by using UV-Visible spectrophotometry (Infinite M 200 Tecan, Switzerland) at 540 nm and the α-amylase inhibitory activity was expressed as percentage inhibition. The assay was carried out in 96-well microplates (Solid clear F-bottom, Greiner Bio One, Austria). The percentage of inhibition was obtained using the following formula: % inhibition = [A reference–(A sample – A blank)/A reference)] ×100.

Regarding α-glucosidase inhibitory activity of carob methanolic extract, it was assessed spectrophotometrically according to Oboh et al. (2012) with few modifications. Briefly, 100 µL phosphate buffer solution of 2.5 U mL−1 α-glucosidase enzyme (Saccharomyces cerevisiae; 2.5 mg 10 mL−1; 10 U mg−1; Sigma-Aldrich. St Louis, USA) and 50 µL of each test sample solution (1 mg mL−1extract) dissolved in 30% DMSO were added in each well and incubated at 25 °C for 10 min. Then, 50 µL of 4-Nitrophenyl (3-D-glucopyranoside (P-NPG) substrate solution (5 mM of P-NPG (Sigma-Aldrich, St Louis, USA) 7.53 mg/5mL dissolved in pH 6.9 phosphate buffer which prepared by mixing with 100 mM monobasic potassium phosphate); was added to the reacting mixture and re-incubated for 5 min at 25°C. Thereafter, 80 µL of 0.1M sodium carbonate solution was added to terminate the catalytic reaction. Acarbose 95% (Acros Organic, USA) dissolved in 30% DMSO was used as a positive control. The reference sample contained the same reaction mixture except the same amount of test sample solution and positive control that were replaced by phosphate buffer solution. The blank was included all reagents and test samples with the exception of α-glucosidase enzyme. All measurements were performed in triplicate. The absorbance of the reaction was recorded on UV-Visible spectrophotometry (Infinite M 200; Tecan, Switzerland) at 405 nm. The assay was carried out in 96-well microplates (Solid clear F-bottom, Greiner Bio One, Austria). The inhibitory rate of samples on α-glucosidase enzyme was obtained using the following formula: % inhibition = [A reference – (A sample – A blank)/A reference)] ×100.

In vitro cytotoxicity of the methanolic extract of carob

Cell culture

RIN-5F (rat pancreatic β-cell line) and WRL 68 (human hepatic cell line) (ATCC; Manassas, VA, USA) were maintained in RPMI-1640 and DMEM medium, respectively. 10% of fetal bovine serum (FBS) and 1% antibiotics (penicillin-streptomycin) were supplemented to the media, then it was incubated under an atmosphere of 5% CO2 and 95% humidified air at 37 °C. The medium was replaced twice weekly until confluent cell monolayer was formed and observed under an inverted microscope.

Cellular viability assay

The inhibitory effect of methanolic extract of carob was determined by MTT assay. The reduction of MTT (3-[4, 5-dimethylthiazol-2-yl]-2, 5-diphenyltetrazolium bromide) by the mitochondrial dehydrogenase in cells and producing purple formazan is the principle of the experiment. In which 5 ×103 of WRL 68 and RIN-5F per well were seeded in triplicate in 96-well microplates and incubated for 24 h to be attached at 37 °C with 5% CO2 saturation. After 24 h of incubation, decreasing concentrations of carob methanolic extract (such as 1,000, 500, 250, 125, 60, 30, 15 and 7 µmol) were prepared (0.025% DMSO) and transferred to the cells with incubating for 24, 48,72 h at 37 °C and 5% CO2. Subsequently, 20 µL of MTT solution (5 mg mL−1) was added to the treated cells in a dark place, covered with aluminum foil and incubated for 4 h. All media was discharged and a total of 100 µL of DMSO was poured into each well until the purple formazan crystals dissolved (Jayash et al., 2016; Jayash et al., 2017). The blank was performed in a similar manner but the test samples were replaced by media (0.025% DMSO). The plates were measured using a microplate reader UV-Visible spectrophotometry (Infinite M 200; Tecan, Switzerland) at absorbance 570 nm. The tests were carried out in 96-well microplates (Solid clear F-bottom; Greiner Bio One, Kremsmuenster, Austria). All experiments were conducted in triplicate to evaluate the IC50 (The concentration of the methanolic carob extract that was able to inhibit 50% of cells viability). The percentage of cell viability was calculated using the following equation: Cell viability% = A treated cells/A blank ×100.

According to Jayash et al. (2017), the cytotoxic effect of each serial dilution of the methanolic extract of carob was evaluated.

Acute oral toxicity of the methanolic extract of carob

Acute oral toxicity was conducted according to the guidelines of OECD (OECD, 2005) and ethical approval (2014-07-01/PHARM/MAQA) that was issued by the Institutional Ethics Committee, University of Malaya, Malaysia. The animals were kept under the standard conditions of housing as mentioned above. Totally, 36 adult SD-rats (18 males and 18 females) were randomly dispersed into normal control (n = 6), low dose methanolic carob extract (CS2000) (n = 6) and high dose methanolic carob extract (CS5000) (n = 6), which received an oral single dose of 5 mL kg−1 normal saline, 2,000 mg kg−1 carob and 5,000 mg kg−1 carob during the 14 days of the acute oral toxicity time course, respectively. Experimental rats were fasted overnight before dosing. Immediately after dosing, rats were observed at half an hour, 2, 4, 24 and 48 h for any toxicological signs. By the 14th day, rats were fasted overnight, anaesthetized, scarified to collect blood samples via cardiac puncture. Blood samples were centrifuged at 1,500 rpm to obtain serum to measure liver function tests, kidney function tests and total protein. as well as liver and kidney were harvested and their sections were processed to be stained with H&E stain for histopathology evaluation.

In vivo glycaemic activity of the methanolic extract of carob

Experimental animals

Totally, 30 male Sprague–Dawley rats (7–8 weeks) were obtained from the Animal House Unit (AEU) in the Faculty of Medicine, University of Malaya. Malaysia. Animals were kept following the Guide for the Care and Use of Laboratory Animals published by (Institute for Laboratory Animal Research, 2010). The use of animals and the study was conducted according to the ethic approval issued by Institutional Animal Care and Use Committee (FOM IACUC), Faculty of Medicine, University of Malay, Malaysia in 18 August 2014 under ethic number (2014-07-01/PHARM/MAQA). The animals were kept within the AEU at Faculty of Medicine, University of Malaya and allowed to acclimatize for one week under 20 ± 3  °C, relative humidity (30–70%) and 12 h light/dark cycle with free access to food and water.

Induction of type 2 diabetic animal model

According to the protocol of (Arya et al., 2012a) , the rats were fasted overnight and injected with a single intraperitoneal dose of 55 mg kg−1 of freshly prepared streptozotocin (STZ) in 0.1 M citrate buffer (pH 4.5). Immediately after 15 mins, rats were injected intraperitoneally with 210 mg kg−1 nicotinamide (NAD) which exert an antioxidant capacity that minimize the cytotoxic actions of STZ and protects pancreatic β-cell against the destructive effect of STZ (Furman, 2015). After 96 h, rats were fasted overnight except free access for tap water to measure fasting blood sugar using Accu-Chek glucometer. The injected rats were monitored for 10 days and those which attained fasting blood sugar between 10-14 mmol L−1 were considered type 2 diabetic.

Dose rationale of carob

In this study, the doses of the methanolic carob extract were selected to be 500 mg kg−1 as a low dose (CS500) and 1,000 mg kg−1 as a high dose (CS1000) according to the pervious study by Rtibi et al. (2015b).

The experimental design

The type2 induced rats were randomly assigned into four groups (n = 6); namely, diabetic untreated control (D0), diabetic glibenclamide-treated (GD), diabetic low dose carob-treated (CS500) and diabetic high dose carob-treated (CS1000), which received a single oral daily dose of 5 mL kg−1 normal saline, 5 mg kg−1 glibenclamide (Hafizur et al., 2015), 500 mg kg−1 carob and 1,000 mg kg−1 carob for four weeks respectively. An extra untreated non-diabetic control (NC) (n = 6) was added and given oral daily single dose of 5 mL kg−1 of normal saline for four weeks. Blood glucose level was monitored weekly from the tail vein using Accu-Chek glucometer. Body weight was recorded at the day of starting (initial body weight) and cession of treatment (final body weight). By the 28th days of treatment, rats were anaesthetized using ketamine (50 mg kg−1) and xylazine (5.0 mg kg−1). Then, 5 mL of blood samples were collected for blood analysis through cardiac puncture. Blood samples were centrifuged at 2,000 rpm for 10 min and serum samples were separated and stored at −80 for the future use. Subsequently, rats were scarified by increasing anesthetic dose (ketamine 150 mg kg−1& xylazine 15.0 mg kg−1) to obtain pancreas for gross pathology examination.

Oral glucose tolerance test

Oral glucose tolerance test (OGTT) evaluated the glucose challenge ability of each animal one week prior to animal sacrifice (Arya et al., 2012b). After fasting rats overnight, tip tail fasting blood glucose was measured at 0 time. Twenty minutes after rats had received their own treatment, rats were given a single oral dose of glucose solution (3 g/kg), and blood glucose was measured at 30, 60, 90 and 120 min.

Evaluation of insulin resistance

After measuring fasting serum insulin, homeostatic model assessment (HOMA-IR, HOMA-β and HOMA-S) was estimated (Mather, 2009). Serum insulin was quantified by the enzyme-linked immunosorbent assay (ELISA) using an ultrasensitive rat insulin ELISA kit (ER1113, Wuhan Fine Biological Technology Co., Ltd., China) and read at 450 nm with a microplate reader (Hydroflex Elisa; Chemo pharm, Vienna, Austria) according to the manufacturer’s instructions. HOMA-IR, HOMA- β and HOMA-S were estimated in term of fasting insulin (µU mL−1) and fasting blood glucose (mmol L−1) using HOMA 2 calculator software (Diabetes Trial Unit UoO, UK, 2004).

Histopathological examinations

For histological analysis, pancreatic tissues were collected directly after sacrificing animals and fixed with formaldehyde 10% (v/v). Histopathological examinations were accomplished by an experienced pathologist. The pancreases sections were stained with hematoxylin and eosin (H&E). Histological features including morphological configuration, size of islets and number of β-cells were examined and differentiated for each group.

Statistical analysis

The statistical analysis was determined using a statistical software package (SPSS for Windows, version 23, IBM Corporation, Armonk, NY, USA) and analysis of variance (ANOVA ), followed by Tukey’s-SHD; multiple range post-hoc test. The IC50 was calculated from the linear regression analysis. Pearson correlation coefficient analysis was done to evaluate the correlation between phenolic and flavonoid contents against antioxidant activities. Values were considered statistically significant at P < 0.05. Results were expressed as either means ± SD or means ± SE.

Results

Antioxidant activity

Table 1 demonstrated that the phenolic content (y = 0.0075x – 0.0123, R2 = 0.9817) was 127.02 ± 7.18 mg GAE g−1, while the flavonoid content (y = 0.0023x – 0.0165, R2 = 0.9979) was 49.74 ± 0.88 mg QE g−1, which means that flavonoid content approximately accounted for 39.2% of the total phenolic content. According to IC50 values, the scavenging activity of the methanolic extract against the free radicals of DPPH (y = 13.339 ln (x) + 17.685, R2 = 0.9727) was close to that of Trolox, but it was double that of quercetin. Similarly, FRAP value (y = 0.0002x + 0.002, R2 = 0.9649) of the methanolic extract was approximately half that of either quercetin or Trolox.

Table 1 Total phenolic and flavonoid with the antioxidant activities of the methanolic extract of Carob.

The table demonstrated that the phenolic content (y = 0.0075x − 0.0123, R2 = 0.9817) was 127.02 ± 7.18 mg GAE g−1, while the flavonoid content (y = 0.0023x − 0.0165, R2 = 0.9979) was 49.74 ± 0.88 mg QE g−1, which means that flavonoid content approximately accounted for 39.2% of the total phenolic content.

	Total phenolic content
(mg GAE g−1)	Total flavonoid Content
(mg QE g−1)	DPPH radical
IC50: µg mL−1	FRAP assay
µmoL FeSo4 mg−1	
Carob extract	127.02 ± 7.18	49.74 ± 0.88	11.23 ± 0.47	3191.67 ± 54.34	
Quercetin	–	–	6.64 ± 0.07	6565.17 ± 678.75	
Trolox	–	–	9.46 ± 0.14	6433 ± 216.49	
Notes.

Values were expressed as mean  ± SE (n = 3).

Pearson correlations analysis displayed that the total phenolic content of carob methanolic extract had a strong positive correlation with FARP assay (r = 0.918, P < 0.01) and a strong negative with DPPH radical (r =  − 0.910, P < 0.01).

Likewise, total flavonoid content of carob methanolic extract reflects a strong positive correlation with FRAP assay (r = 0.886, P < 0.01) and a strong negative correlation with DPPH radical (r =  − 0.867, P < 0.01), (Table 2).

Figure 2 Effect of the methanolic extract of carob on viability of pancreatic RIN-5F cells.

The percentage of viability of RIN-5F cells after 24, 48 and 72 h of exposure to the methanolic extract of carob. The percentage of viability was upper than 30%. Values were expressed as mean ±SD (n = 3).

Table 2 Correlation coefficients of phenolic and flavonoid contents against antioxidant assays.

The Pearson correlations analysis displayed that the total phenolic content of carob methanolic extract had a strong positive correlation with FARP assay (r = 0.918, P < 0.01) and a strong negative with DPPH radical (r =  − 0.910, P < 0.01). Likewise, total flavonoid content of carob methanolic extract reflects a strong positive correlation with FRAP asaay (r = 0.886, P < 0.01) and a strong negative correlation with DPPH radical (r =  − 0.867, P < 0.01).

Methanolic carob extract	FRAP assay	DPPH radical	
Total phenolic content	0.918**	−0.910**	
Total flavonoid content	0.886**	−0.867**	
Notes.

** means correlation is significant at the 0.01 level (2-tailed).

Figure 3 Effect of the methanolic extract of carob on viability of WRL 68 cells.

The percentage of viability of WRL 68 cells after 24, 48 and 72 h of exposure to the methanolic extract of carob. The percentage of viability was upper than 30%. Values were expressed as mean ±SD (n = 3).

Toxicity assessment of the methanolic extract of carob

In vitro cytotoxicity

In the current study, the toxic effect of the methanolic extract of carob against RIN-5F (pancreatic β-cells) and WRL 68 (human hepatocytes cells) was assessed. The percentage of viability of either RIN-5F (Fig. 2) or WRL 68 (Fig. 3) was upper than 30% after 24, 48 and 72 h of exposure to the different serial concentrations of the methanolic extract of carob.

Table 3 Body weight as well as absolute and relative organ weight of female and male rats.

No significant difference was observed in body weight of male and female rats. Like body weight, absolute liver, kidney, heart and pancreas weights, as well as relative liver, kidney, heart and pancreas weights of male and female rats were not significantly different.

	Females	Males	
	NC	CS2000	CS5000	NC	CS2000	CS5000	
BW0	230.00 ± 15.16	226.67 ± 18.61	230.83 ± 13.93	356.67 ± 22.73	363.33 ± 39.33	360.83 ± 36.66	
BW1	246.33 ± 16.08	251.67 ± 24.83	260.83 ± 14.63	410.00 ± 24.70	407.50 ± 41.32	402.50 ± 37.52	
BW2	250.00 ± 20.73	244.17 ± 21.77	237.50 ± 15.73	399.17 ± 25.3	406.67 ± 39.96	391.67 ± 40.21	
ALW	7.88 ± 0.75	7.53 ± 0.86	7.40 ± 1.12	12.70 ± 0.83	12.23 ± 1.72	12.2 ± 1.1	
AKW	1.78 ± 0.25	1.55 ± 0.19	1.65 ± 0.19	2.77 ± 0.23	2.65 ± 0.31	2.70 ± 0.36	
AHW	0.85 ± 0.15	0.85 ± 0.10	0.80 ± 0.09	1.13 ± 0.16	1.35 ± 0.19	1.22 ± 0.15	
APW	1.22 ± 0.15	1.15 ± 0.19	1.05 ± 0.24	1.47 ± 0.29	2.00 ± 0.59	1.78 ± 0.50	
RLW	0.032 ± 0.001	0.031 ± 0.002	0.031 ± 0.003	0.032 ± 0.002	0.030 ± 0.002	0.031 ± 0.001	
RKW	0.007 ± 0.001	0.006 ± 0.0003	0.007 ± 0.001	0.007 ± 0.001	0.007 ± 0.0003	0.007 ± 0.0004	
RHW	0.003 ± 0.0005	0.004 ± 0.0003	0.003 ± 0.0003	0.003 ± 0.0004	0.003 ± 0.0003	0.003 ± 0.0002	
RPW	0.005 ± 0.0003	0.005 ± 0.0005	0.004 ± 0.0007	0.004 ± 0.001	0.005 ± 0.001	0.004 ± 0.001	
Notes.

NC normal control; CS2000, 2,000 mg kg−1 carob-treated; CS5000, 5,000 mg kg−1 carob-treated; BW0, 1 and 2, body weight before treatment as well as at the end of 1st and 2nd week of treatment, respectively

ALW absolute liver weight

AKW absolute kidney weight

AHW absolute heart weight

APW absolute pancreas weight

RLW relative liver weight

RKW relative kidney weight

RHW relative heart weight

RPW relative pancreas weight. Values are expressed as mean  ±SD, (n = 6)

In vivo acute oral toxicity

Neither morbidity nor mortality in male or female rats was observed. Similarly, no abnormal signs or behavioral changes were observed during the course of acute oral toxicity. Table 3 showed that no significant difference was observed in body weight of male and female rats. Like body weight, absolute liver, kidney, heart and pancreas weights, as well as relative liver, kidney, heart and pancreas weights of male and female rats were non-significantly different. Table 4 showed that only alkaline phosphatase and urea of high dose carob-treated (CS5000) were significantly increased as compared to normal control (NC) (P ≤ 0.05 and P ≤ 0.01, respectively). In female rats, only serum creatinine showed significant increased as compared to normal control (P ≤ 0.01).

Table 4 Biochemical parameters for male and female rats.

Only alkaline phosphatase and urea of CS5000 were significantly increased as compared to normal control (NC). In female rats, only serum creatinine showed significant increased as compared to noramal control.

	Females	Males	
	NC	CS2000	CS5000	NC	CS2000	CS5000	
ALP	73.60 ± 3.21	76.20 ± 20.99	93.60 ± 16.13	76.80 ± 11.97	79.00 ± 12.25	142.00 ± 7.87**	
AST	93.80 ± 10.03	126.80 ± 32.58	132.60 ± 38.49	101.60 ± 13.46	121.20 ± 14.57	100.60 ± 25.77	
ALT	59.80 ± 4.82	53.20 ± 11.90	57.80 ± 1.64	34.00 ± 4.74	34.80 ± 3.77	40.20 ± 5.93	
TP	30.00 ± 1.87	31.60 ± 7.02	35.20 ± 2.86	7.26 ± 0.42	7.28 ± 0.38	7.06 ± 0.46	
Cr	6.12 ± 0.52	7.36 ± 1.68	9.62 ± 1.93**	51.80 ± 6.98	53.20 ± 2.68	52.00 ± 2.55	
Ur	7.58 ± 0.26	7.58 ± 0.57	7.00 ± 0.53	7.06 ± 1.21	7.36 ± 1.07	11.20 ± 1.18*	
Notes.

NC normal control; CS2000, 2,000 mg kg−1 carob-treated; CS5000, 5,000 mg kg−1 carob-treated

ALP alkaline phosphatase

AST aspartate aminotransferase

ALT alanine aminotransferase

TB total bilirubin

Cr creatinine

Ur urea

*, ** symbolized a significant difference at either P < 0.05 or P ≤ 0.01 versus normal control. Values were expressed as mean ±SD, (n = 5).

Histopathology of acute oral toxicity

The liver histological sections of normal control (G and J), low dose carob-treated CS2000 (H and K) and high dose carob-treated CS5000 (I and L) of both male and female were of normal architecture of hepatocytes at the level of their cell membrane, nuclei or even cytoplasm. In addition, no congestion in portal veins, no necrotic lesions, no inflammatory signs or fatty infiltrations could be observed. Similarly, kidney histological sections of normal control (A and D), CS2000 (B and E) and CS5000 (C and F) of both male and female were of normal architecture of tubules, Bowman’s capsule, Malpighian corpuscles (Fig. 4).

Figure 4 Hematoxylin and eosin stained histopathological sections of livers and kidneys of male and female SD-rats.

The hematoxylin and eosin stained histopathological sections of livers and kidneys of male and female SD-rats. (A–C) normal control (NC): 2,000 mg kg−1 carob-treated (CS2000) and 5,000 mg kg−1 carob-treated (CS5000) male kidney sections, respectively. (D–F) normal control (NC): 2,000 mg kg−1 carob-treated (CS2000) and 5,000 mg kg−1 carob-treated (CS5000) female kidney sections, respectively. (G–I) normal control (NC): 2,000 mg kg−1 carob-treated (CS2000) and 5,000 mg kg−1 carob-treated (CS5000) male liver sections, respectively. (J–L) normal control (NC): 2,000 mg kg−1 carob-treated (CS2000) and 5,000 mg kg −1 carob-treated (CS5000) female liver sections, respectively. Liver sections of (G–L) showed similar normal architecture of hepatocytes at the level of their cell membrane, nuclei or even cytoplasm without any abnormalities such as congestion of portal veins, necrotic lesions, inflammatory signs or fatty infiltrations. Similarly, kidney sections of all groups similar normal architecture of tubules, Bowman’s capsule, Malpighian corpuscles without any abnormal findings (magnification × 40).

Glycemic effect of the methanolic extract of carob

In vitro glycemic effect

The IC50 of carob methanolic extract (92.99 ± 0.22 µg mL−1) was higher than that of α-acarbose (23.33 ± 0.73μg mL−1) against α-amylase (Fig. 5). Similarly, the IC50 of the methanolic extract (97.13 ± 4.11μg mL−1) against α-glucosidase was higher than that of α-acarbose (27.05 ± 0.99μg mL−1) (Fig. 6).

Figure 5 Inhibitory effect of the methanolic extract of carob against α-amylase.

The percentage of inhibition of the methanolic carob extract and α-acarbose against against α-amylase. The IC50 of carob methanolic extract (92.99 ± 0.22 µg mL−1) was higher than that of α-acarbose (23.33 ± 0.73 µg mL−1) against α-amylase.

Figure 6 Inhibitory effect of the methanolic extract of carob against α-glucosidase.

The percentage of inhibition of the methanolic carob extract and α-acarbose against α-glucosidase. The IC50 of the methanolic carob extract (97.13 ± 4.11 µg mL−1) against α-glucosidase was higher than that of α-acarbose (27.05  ± 0.99 µg mL−1).

In vivo glycemic effect

Body weights as well as absolute and relative organs weight of STZ-NAD diabetic animal model

Table 5 demonstrates that body weight of low dose carob-treated (CS500) and high dose carob-treated (CS1000) was significantly reduced more than untreated type 2 diabetic rats (D0) (P ≤ 0.05 and P ≤ 0.01, respectively). On the other hand, absolute pancreas weight (APW) and relative pancreas weight (RPW) of CS1000 were significantly lower than that of untreated type 2 diabetic rats group (P ≤ 0.001 and P ≤ 0.01, respectively).

Table 5 Body weight as well as absolute and relative pancreas weight of STZ-NAD diabetic animal model.

The table demonstrated that body weight of CS500 and CS1000 was significantly reduced more than untreated type 2 diabetic rats (D0). On the other hand, absolute pancreas weight (APW) and relative pancreas weight (RPW) of CS1000 were significantly lower than that of untreated type 2 diabetic rat group.

	NC	D0	GD	CS500	CS1000	
BW0	287.2 ± 32.31	289.2 ± 17.42	282.3 ± 22.84	273.3 ± 16.33	288.7 ± 20.71	
BW4	360.3 ± 30.92	333.7 ± 14.53	306.0 ± 8.31	296.8 ± 13.82*	281.3 ± 25.02**	
APW	2.34 ± 0.16	2.02 ± 0.33	1.79 ± 0.14	1.62 ± 0.36	1.27 ± 0.37***	
RPW	0.0066 ± 0.001	0.0063 ± 0.001	0.0059 ± 0.001	0.0055 ± 0.001	0.0044 ± 0.001**	
Notes.

NC normoglycemic untreated rats; D0, untreated type 2 diabetic rats; GD, glibenclamide-treated type 2 diabetic; CS500, 500 mg kg−1 of carob-treated type 2 diabetic; CS1000, 1,000 mg kg −1 of carob-treated type 2 diabetic; BW0 and 4, initial body and final body weight, respectively

APW absolute pancreas weight

RPW relative pancreas weight

*, ***, *** denoted a significant difference at P ≤ 0.05, P ≤ 0.01 and P ≤ 0.001 versus D0. Values were expressed as mean ±SE, (n = 6).

Weekly fasting blood sugar of STZ-NAD diabetic animal model

Figure 7 showed that normal control (NC) showed a higher significant fasting blood glucose (FBG) than untreated type 2 diabetic rats (D0) at the baseline as well as the end of the 1st, 2nd, 3rd and 4th week of treatment (P ≤ 0.001, P ≤ 0.001, P ≤ 0.001, P ≤ 0.001 and P ≤ 0.001, respectively). In addition, FBG of glibenclamide-treated type 2 diabetic (GD) was significantly lower than untreated type 2 diabetic rats at the end of the 2nd, 3rd, and 4th weeks (P ≤ 0.001, P ≤ 0.001 and P ≤ 0.001, respectively). Moreover, low dose carob-treated (CS500) showed a lower significant FBG than untreated type 2 diabetic rats only at the end of 4th week (P ≤ 0.001) while high dose carob-treated (CS1000) exhibited a lower significant FBG than untreated type 2 diabetic rats at the end of 3rd and 4th weeks (P ≤ 0.01 and P ≤ 0.001, respectively).

Figure 7 Weekly fasting blood glucose of STZ-NAD diabetic animal model.

The fasting blood glucose (FBG) of normal control group (NC) was higher significant than untreated type 2 diabetic rats (D0) at the baseline as well as the end of the 1st, 2nd, 3rd and 4th week of treatment. In addition, FBG of glibenclamide-treated type 2 diabetic (GD) was significantly lower than D0 at the end of the 2nd, 3rd, and 4th weeks. Moreover, CS500 showed a lower significant FBG than D0 only at the end of 4th week while CS1000 exhibited a lower significant FBG than D0 at the end of 3rd and 4th weeks. NC: normoglycemic untreated rats, D0: untreated type 2 diabetic, GD: glibenclamide-treated type 2 diabetic, CS500: 500 mg kg−1 of carob-treated type 2 diabetic, CS1000: 1,000 mg kg−1 of carob-treated type 2 diabetic. FBG_0, 1, 2, 3 and 4: fasting blood glucose at baseline as well as at the end of the 1st, 2nd, 3rd and 4th week of treatment, respectively. Superscript ###: denoted a significant difference at P ≤ 0.001 versus normal control. Superscript **, ***: denoted a significant difference at P ≤ 0.01 and P ≤ 0.001 versus D0. Values were expressed as mean ±SE, (n = 6).

Oral glucose tolerance test (OGTT)

Figure 8 illustrates that blood glucose of untreated type 2 diabetic (D0) was significantly higher than that of normal control (NC) at zero (P ≤ 0.01), 30 (P ≤ 0.001), 60 (P ≤ 0.001), 90 (P ≤ 0.001) and 120 (P ≤ 0.001). Conversely, blood glucose of glibenclamide-treated type 2 diabetic (GD) was significantly lower than that of untreated type 2 diabetic at 30 (P ≤ 0.05), 60 (P ≤ 0.001), 90 (P ≤ 0.001) and 120 (P ≤ 0.001). However, blood glucose of low dose carob-treated (CS500) and high dose carob-treated (CS1000) was significantly lower than that of untreated type 2 diabetic at 90 (P ≤ 0.01 and P ≤ 0.001, respectively) and 120 min (P ≤ 0.001 and P ≤ 0.001, respectively).

Figure 8 Oral glucose tolerance test of STZ-NAD diabetic animal model.

The blood glucose of untreated type 2 diabetic rats (D0) was significantly higher than that of normal control (NC) at zero, 30, 60, 90 and 120. Conversely, blood glucose of glibenclamide-treated type 2 diabetic (GD) was significantly lower than that of untreated type 2 diabetic at 30, 60, 90 and 120. However, blood glucose of CS500 and CS1000 was significantly lower than that of untreated type 2 diabetic at 90 and 120 minutes. NC: normoglycemic untreated rats, D0: untreated type 2 diabetic, GD: glibenclamide-treated type 2 diabetic, CS500: mg kg−1 of carob-treated type 2 diabetic, CS1000:1000 mg kg−1 of carob-treated type 2 diabetic. Superscript ##, ###: denoted a significant difference at P ≤ 0.01 and P ≤ 0.001 versus normal control. Superscript *, **, ***: denoted a significant difference at P ≤ 0.05, P ≤ 0.01 and P ≤ 0.001 versus D0. Values were expressed as mean ±SE, (n = 6).

Biochemistry and hemostatic assessment index of STZ-NAD diabetic animal model

Table 6 showed that total protein of glibenclamide-treated type 2 diabetic (GD) and high dose carob-treated (CS1000) was significantly higher than that of untreated type 2 diabetic (D0) (P ≤ 0.01 and P ≤ 0.01, respectively). On the other hand, serum amylase of untreated type 2 diabetic was significantly lower than that of normal control (NC) (P ≤ 0.001), glibenclamide-treated type 2 diabetic ( P ≤ 0.001), low dose carob-treated (CS500) (P ≤ 0.001) and high dose carob-treated (P ≤ 0.001). Serum FBG of untreated type 2 diabetic was higher than that of normal control (P ≤ 0.001), while serum FBG of glibenclamide-treated type 2 diabetic and high dose carob-treated was significantly lower than that of untreated type 2 diabetic (P ≤ 0.001 and P ≤ 0.001, respectively). However, only insulin of untreated type 2 diabetic was less than that of normal control (P ≤ 0.001). In addition, HOMA-B of untreated type 2 diabetic was significantly less than that of normal control (P ≤ 0.001), glibenclamide-treated type 2 diabetic (P ≤ 0.001), low dose carob-treated (P ≤ 0.05) and high dose carob-treated (P ≤ 0.01), while HOMA-S and HOMA-IR showed a non-significant difference.

Table 6 Biochemistry and hemostatic assessment index of STZ-NAD diabetic animal model.

The total protein of glibenclamide-treated type 2 diabetic (GD) and CS1000 was significantly higher than that of untreated type 2 diabetic (D0). On the other hand, serum amylase of untreated type 2 diabetic was significantly lower than that of normal control, glibenclamide-treated type 2 diabetic, CS500 and CS1000. Serum FBG of untreated type 2 diabetic was higher than that of normal control (NC), while serum FBG of glibenclamide-treated type 2 diabetic and CS1000 was significantly lower than that of untreated type 2 diabetic. However, only insulin of untreated type 2 diabetic was less than that of normal control. In addition, HOMA-B of untreated type 2 diabetic was significantly less than that of normal control, glibenclamide-treated type 2 diabetic, CS500 and CS1000, while HOMA-S and HOMA-IR showed a non-significant difference.

	NC	D0	GD	CS500	CS1000	
TP g L−1	90.60 ± 6.40	77.10 ± 12.71	100.52 ± 7.06**	86.68 ± 10.37	101.16 ± 8.65*	
Amy U L−1	2,891.80 ± 22.1	1,717.00  ±24.81###	2,757.60 ± 18.91***	2,863.00 ± 21.20***	2,910.60 ± 22.42***	
TG mmol L−1	0.52 ± 0.24	0.63 ± 0.41	0.81 ± 0.27	0.77 ± 0.24	0.84 ± 0.32	
TC mmol L−1	2.70 ± 0.24	2.52 ± 1.07	3.16 ± 0.30	2.82 ± 0.49	3.36 ± 0.69	
FBG mmol L−1	4.90 ± 0.41	13.51  ±1.23###	6.50 ± 0.40***	12.34 ± 0.90	10.35 ± 0.85***	
IN mU L−1	15.55 ± 1.26	10.60  ±0.95###	13.15 ± 2.22	11.26 ± 1.25	12.44 ± 1.87	
HOMA- β	163.28 ± 24.46	17.43  ±0.93###	82.48 ± 12.41***	22.85 ± 2.95*	33.88 ± 7.39**	
HOMA-S	51.12 ± 4.32	55.73 ± 5.89	56.80 ± 8.33	56.43 ± 7.42	54.03 ± 7.49	
HOMA-IR	1.97 ± 0.19	1.78 ± 0.22	1.80 ± 0.31	1.80 ± 0.23	1.88 ± 0.26	
Notes.

NC normoglycemic untreated rats; D0, untreated type 2 diabetic; GD, glibenclamide-treated type 2 diabetic; CS500, 500 mg kg−1 of carob-treated type 2 diabetic; CS1000, 1,000 mg kg−1 of carob-treated type 2 diabetic

TP total protein

Amy amylase

FBG fasting blood glucose

IN insulin

HOMA- β β eta cell function

HOMA-S insulin sensitivity

HOMA-IR insulin resistance

### denoted a significant difference at P ≤ 0.001 versus normal control.

*, **, *** denoted a significant difference at P ≤ 0.05, P ≤ 0.01 and P ≤ 0.001 versus D0. Values were expressed as mean ±SD, (n = 6).

Histopathology of STZ-NAD diabetic animal model

Figure 9 showed that histological examination of pancreatic tissues exhibited that normoglycemic untreated rats (A) showed a normal tissue morphological configuration and a larger islet (indicated by red arrow) with high number of β-cells (indicated by yellow arrow), while the untreated type 2 diabetic (B) had the smallest islet and structural malformation with lowest number of β-cells. The high dose carob-treated (E), low dose carob-treated (D) and glibenclamide-treated type 2 diabetic (C) groups had comparatively larger islets with some abnormal structure and higher numbers of β-cells were observed as compared to the untreated type 2 diabetic rats (B).

Figure 9 Hematoxylin and eosin stained nicotinamide streptozotocin-induced diabetic rats.

The histological examination of pancreatic tissues exhibited that normoglycemic untreated rats (A) showed a normal tissue morphological configuration and a larger islet (indicated by red arrow) with high number of β-cells (indicated by yellow arrow), while the untreated type 2 diabetic (B) had the smallest islet and structural malformation with lowest number of β-cells. In addition, high dose carob-treated type 2 diabetic CS1000 (E), low dose carob-treated type 2 diabetic CS500 (D) and glibenclamide-treated type 2 diabetic (C) had comparatively larger islets with some abnormal structure and higher numbers of β-cells were observed as compared to the untreated type 2 diabetic (B). (A) normoglycemic untreated rats (NC), (B) untreated type 2 diabetic (D0), (C) glibenclamide-treated type 2 diabetic (GD), (D) 500 mg kg−1 of carob-treated type 2 diabetic (CS500), (E):1,000 mg kg−1 of carob-treated type 2 diabetic (CS1000). Red and yellow arrows denoted to pancreatic islets and β-cells, respectively (magnification ×40).

Discussion

The extraction solvents play a vital role in the pharmacological activities (Turkmen, Sari & Velioglu, 2006) since the content of an extract is almost dependent on the extraction solvent, therefore, the current study used methanol as a solvent of extraction for carob because it is one of the universal solvents which is capable of extracting non- polar, semi-polar and polar constituents (Saghir et al., 2016).

Earlier reports indicated that carob pods contain considerable amounts of phenolic compounds such as tannins (proanthocyanidins, gallotannins and ellagitannins), flavonoids, gallic acid, (+)-catechin, (−)-epigallocatechin gallate, cinnamic acid, p-coumaric acid and quercetin glycosides (Gulay et al., 2012; Marakis, 1996; Owen et al., 2003; Rtibi et al., 2017a). Our findings revealed that the methanolic extract of carob had a high content of total phenolic and flavonoids. In the current study, the antioxidant activity of the methanolic carob extract using DPPH and FRAP assays was evaluated prior to conduct its in vivo antioxidant activity to provide a preliminary evidence about the antioxidant activity of the Yemeni cultivar of carob which has not been yet reported. Actually, our findings of in vitro antioxidant activity indicated that the methanolic extract of carob produced a scavenging activity against free radicals of DPPH which was higher than that was reported by Rtibi et al. (2017b). Likewise, the methanolic extract of carob exhibited a ferric reducing antioxidant power that accounted for 50% of those exhibited by quercetin and Trolox. The antioxidant properties of the methanolic extract of carob maybe act through its ability to neutralize reactive oxygen species and myeloperoxidase (Rtibi et al., 2017a; Rtibi et al., 2017b). Nonetheless, the in vitro and in vivo antioxidant activity for most plants was not always the same (Kasote et al., 2015). Accordingly, further studies should consider the effect of methanolic extract of carob on the in vivo antioxidant enzymatic system using pancreatic homogenate.

Universally, natural products are considered as one of the most common source of drugs (Newman, Cragg & Snader, 2003). Regardless of the effectiveness of natural products, those products should firstly be assessed for their safety (Evans et al., 2001; Mounanga, Mewono & Angone, 2015). In the present study, the safety of the methanolic extract of carob was assessed in vitro on pancreatic β-cells and hepatocytes followed by in vivo assessment of its acute toxicity on male and female rats for 14 days. Accordingly, our findings indicated that the methanolic extract of carob was not toxic against pancreatic β cells and hepatocytes as long as the viability of both cells remained more than 30% after 24, 48 and 72 h of exposure to carob (Jayash et al., 2017). Although in vivo acute oral toxicity findings indicated a rise in ALP and urea of male rats as well as creatinine of female rats. As it is known that liver is the main site of biotransformation of xenobiotics into metabolites which could sometimes be hepatotoxic (Sturgill & Lambert, 1997) inducing a rise in hepatic biomarkers (ALP, AST and ALT) and histological abnormalities through destructing cell membrane and structures of hepatocytes (Apaydin et al., 2017). However, ALP is ubiquitous in several body organs and has several isoenzymes (Moss, 1982). Accordingly, the increase in ALP of male rats could be extrahepatic. Regarding the increase in creatinine of female rats, it is well-known that creatinine is a measure of kidney functions, however, it was reported that creatinine is no longer imprecise biomarker (Inker et al., 2012). Consequently, this study relied on the histological findings in livers and kidneys sections of male and female rats to detect any direct toxicity evidence due to the exposure to carob. Actaully, the hsitological sections of liver of male and female rats showed normal architecture of hepatocytes (at the level of their cell membrane, nuclei or even cytoplasm), without necrotic lesions, inflammatory signs or fatty infiltrations. Similarly, histological sections of kidneys of male and female rats showed normal architecture of tubules, Bowman’s capsule, Malpighian corpuscles.

Carob as a plant bears a great significance in the traditional medicine because of its various pharmacological activities (Gulay et al., 2012; Rtibi et al., 2017a). Interesetingly, carob is a native plant to Yemen and it is used as a traditional medicine to improve glycemic control in diabetics long time ago (Battle & Tous, 1997; Papakonstantinou et al., 2017). In point of fact, the previous reports suggested that the physiological viscous property of soluble fiber of carob pods gum could modify carbohydrates structure during digestion which can modify the rate of carbohydrate degradation and control blood glucose levels (Barak & Mudgil, 2014; Forestieri et al., 1989; Marles & Farnsworth, 1995). Consequently, this drew our attention to examine the in vitro inhibitory effect of carob against α-amylase and α-glucosidase which are responsible for post-prandial conversion of carbohydrates into glucose units (Etxeberria et al., 2012). Accordingly, to assess the glycemic activity of the methanolic extract of carob, its in vitro glycemic activity was evaluated in term of its inhibitory effect on α-amylase and α-glucosidase (Etxeberria et al., 2012). Our findings indicated that carob could inhibit those enzymes, however, its IC50 against both enzymes was higher than that of α-acarbose indicating that carob has in vitro hypoglycemic activity but it was not so effective as α-acarbose. On the other hand, the in vivo glycemic activity of such extract was assessed in T2D animal model. Actually, SD-rats were selected to create a type 2 diabetic animal model (T2D) since those rats are most commonly used to investigate the glycemic effects of medicinal plants (Furman, 2015). In addition, SD-rats are more susceptible to become diabetic with small dose of streptozotocin (STZ), and its ability to produce a dose-dependent destruction to pancreatic β-cells (Abeeleh et al., 2009; Furman, 2015). Nevertheless, several protocols have been adopted to induce type 2 diabetes mellitus (T2DM). In this study, animals were injected with intraperitoneal dose of 55 mg kg−1 of STZ followed by an intraperitoneal dose of 210 mg kg−1 of nicotinamide, which prevented STZ from producing a complete destruction of pancreatic β-cells in order to create T2D animal model of relative insulin secretion (Furman, 2015). Our findings indicated that STZ-NAD induced rats showed FBG of 12.74 ± 0.99 mmol L−1that fall within the proposed range of FBG of T2DM (7.2–15.6 mmol L−1) (Furman, 2015; Zhang et al., 2009). Such level of FBG was maintained through the time course of treatment indicating that the created T2D animal model was stable.

Blood glucose of T2D rats after oral glucose challenge indicated that both low dose carob-treated (CS500) and high dose carob-treated (CS1000) showed an antihyperglycemic effect after 90 and 120 min. However, only high dose carob-treated exerted a significant reduction in FBG at the end of the 3rd and 4th week of treatment, which was emphasized by the significant increase of total protein of high dose carob-treated due to alleviating the catabolic effect of the persistent hyperglycemia on proteins (Ng, Ton & Kadir, 2016), and the significant reduction in body weigh as a sign of the resulted hypoglycemia (Pandit, Phadke & Jagtap, 2010). The resulted hypoglycemic effect of high dose carob-treated might be due to the low GIT value of carob and its high content of fibers which provoked feeling of satiety (Papakonstantinou et al., 2017) or the polyphenols content of carob could chelate sugars, lipids and fibers leading to reducing their intestinal absorption (Lattimer & Haub, 2010; Williamson, 2013).

In normoglycemic control rats, the insulin level was significantly higher than that of untreated type 2 diabetic group (D0). Conversely, the insulin level of all treated and untreated diabetic rats was non-significantly different, although it was suggested that carob could enhance insulin release (Forestieri et al., 1989; Marles & Farnsworth, 1995). For other in vivo mechanisms and pathways, our study recommended further investigation on the effect of carob on GLP-1. On the other hand, HOMA-B, as surrogate of pancreatic β-cells activity (Wallace, Levy & Matthews, 2004) was significantly higher than that of untreated type 2 diabetic due to that normoglycemic rats still had intact pancreatic β-cells since they were not exposed to the destructive effect of streptozotocin (Furman, 2015), which was emphasized with their normal findings of the pancreatic histological sections. Similarly, HOMA-S and HOMA-IR of all treated and untreated diabetic rats as well as normal control showed a non-significant difference. Actually, the carob pod is rich in D-pinitol compound (Tetik et al., 2011) which could play an insulin-like role in improving the insulin sensitivity (Gao et al., 2015). Nonetheless, the used model was streptozotocin-nicotinamide diabetes induced (insulin deficient model) rather than insulin resistant induced model (Furman, 2015). On the other hand, the slight increase of insulin in of glibenclamide-treated type 2 diabetic (GD) group was due to that glibenclamide is a well-known insulin tropic agent (Chan & Colagiuri, 2015), while the slight increase in insulin of high dose carob-treated (CS1000) could be independent of insulin (Papakonstantinou et al., 2017). Perhaps carob provided a protection to the remaining β-cell from destruction owing to its antioxidant activity (Coskun et al., 2005; Kaneto et al., 1999; Nasri et al., 2015) particularly that our findings of the scavenging activity of carob against free radicals of DPPH was indicated. As mentioned above, however, the findings of the in vitro antioxidant activity of carob extract could not be extrapolated to reflect the in vivo antioxidant activity (Kasote et al., 2015). Nonetheless, low dose carob-treated and high dose carob-treated showed a higher significant value of HOMA-B than that of untreated type 2 diabetic (D0) indicating that low dose carob-treated (CS500) and high dose carob-treated (CS1000) still had active insulin secreting β-cells, which could be supported by higher number of β-cells that was observed in the histological pancreatic sections of low dose carob-treated (D) and high dose carob-treated (E) as compared to untreated type 2 diabetic (B).

Conclusion

Carob pods did not cause acute systemic toxicity and showed in vitro antioxidant effects. On the other hand, the in vitro ex-pancreatic hypoglycemic effect of the methanolic extract of carob pods was evident through inhibiting α-amylase and α-glucosidase. Interestingly, a high dosage of carob exhibits in vivo antihyperglycemic activity and warrants further in-depth study to identify the bioactive constituents in the potential methanolic extract of carob pods.

Supplemental Information

Data S1 Raw data of total phenolic content and total flavonoid content

The excel file contains calculations of total phenolic content and total flavonoid content as well as positive controls quercetin and gallic acid.

Click here for additional data file.

Data S2 Raw data of biological replicate of DPPH assay

The dataset contains three technical replicates of measuring the ability to scavenge DPPH radical by methanolic carob extract as well as positive controls (quercetin and trolox).

Click here for additional data file.

Data S3 Raw data of ferric reducing antioxidant power (FRAP)

The dataset contains raw triplicates absorbance of methanolic carob extract, trolox and quercetin against FRAP. Ferrous sulphate standard absorbance is inculded.

Click here for additional data file.

Data S4 Raw data of biological replicates of viability of rat pancreatic β-cell RIN-5F and human hepatic WRL 68 cells

The dataset contains the biological replicates absorbance of viability of RIN-5F and WRL 68 cells that treated with serial concentrations of methanolic carob extract for 24, 48 and 72 hours.

Click here for additional data file.

Data S5 Raw data of acute oral toxicity study

The dataset involves body weight of male and female rats with absolute and relative organs weight within two weeks, as well as biochemical parameters for both male and female rats.

Click here for additional data file.

Data S6 Raw data of percentage of inhibition of α-amylase and and α-glucosidase inhibitory

The dataset includes the absorbance of inhibition of methanolic carob extract and α-acarbose as a positive control against α-amylase and α-glucosidase.

Click here for additional data file.

Data S7 Raw data of in vivo type 2 diabetic rats

The raw data contains in vivo type 2 diabetic rat parameters. The dataset includes body weight as well as absolute and relative pancreas weight, biochemistry parameters, hemostatic assessment index, weekly fasting blood sugar and oral glucose tolerance test.

Click here for additional data file.

Additional Information and Declarations

Competing Interests

Author Contributions

Animal Ethics

Data Availability

The authors declare there are no competing interests.

Mousa A. Qasem conceived and designed the experiments, performed the experiments, analyzed the data, prepared figures and/or tables, authored or reviewed drafts of the paper.

Mohamed Ibrahim Noordin and Aditya Arya conceived and designed the experiments, contributed reagents/materials/analysis tools, authored or reviewed drafts of the paper, approved the final draft.

Abdulsamad Alsalahi analyzed the data.

Soher Nagi Jayash cytotoxicity.

The following information was supplied relating to ethical approvals (i.e., approving body and any reference numbers):

The use of animals and the study was conducted according to the ethic approval issued by Institutional Animal Care and Use Committee (FOM IACUC), Faculty of Medicine, University of Malay, Malaysia in 7 June 2014 under ethics reference number (2014-07-01/PHARM/MAQA).

The following information was supplied regarding data availability:

The raw data are provided in the Supplemental Files.

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
