# Peer review of "Evaluation of the glycemic effect of Ceratonia siliqua pods (Carob) on a streptozotocin-nicotinamide induced diabetic rat model"

_PeerJ, doi:10.7717/peerj.4788_

## Round 0.1 · original submission · Minor Revisions

I have received the reviews on your manuscript, "Evaluation of the glycemic effect of Ceratonia siliqua pods (Carob) on nicotinamide-streptozotocin-induced diabetic rats model ", which you submitted to PeerJ.

Based on the advice received, I have decided that your manuscript could be reconsidered for publication should you be prepared to incorporate minor revisions. However, we are not prepared to accept your manuscript in its present form. When preparing your revised manuscript, you are asked to carefully consider the reviewer comments, which are attached, and submit a list of responses to the comments. Your list of responses should be uploaded as a file in addition to your revised manuscript.

Reviewer 1 ·

Basic reporting

There are some minor details that should be reviewed:
- FeSO4•7H2O subscript (P.11 L. 139)
- IC50 defined at the first use
- Uniform fonts and font sizes should be used. Please, review references and legends.

Experimental design

No Comment

Validity of the findings

Point 1. Please explain why the authors chose the doses for the treatments (500 mg kg-1 carob and 1000 mg kg-1 carob) in the induction of type 2 diabetic animal model? This point should be includes in materials and methods, would not be interesting a dose course?
Point 2. The authors describe the Free radical scavenging capacity and Ferric reducing antioxidant power as markers antioxidant activity In fact, Free radical scavenging capacity and FRAP can be used with this goal. However, it is not the only methodology, for exemple Lipid peroxidation (MDA levels) is well know used for antioxidante activity. Why the authors opted for DPPH and FRAP?
Point 3. ‘’...Although in vivo acute oral toxicity findings indicated a rise in ALP and urea of male rats as well as creatinine of female rats, the hsitological sections of liver of male and female rats showed normal architecture of hepatocytes... ‘’ What the authors think about this?
Point 4. HOMA-S and HOMA-IR showed a non-significant difference. Please
please includes a discussion for these results.
Point 5. The study shows in vitro antioxidant activity in assay with synthetic radical (DPPH), that does not mimic physiological conditions. But in the discussion section the authors atributed the exvivo protection to the remaining β-cell from destruction owing to its antioxidant activity. What the authors think about this? Also, authors concludes that Carob pods possess antioxidant effects by inhibiting α-amylase and α-glucosidase. In my point of view the authors should explain this conclusion.
Point 6. Authors concludes that ‘’Carob pods are non-toxic’’, it will be better: did not cause acute systemic toxicity. The conclusion section should be rewritten.

Additional comments

This manuscript describes the antioxidante and antihyperglycemic activity of the methanolic extract of Carob in nicotinamide-streptozotocin-induced diabetic rat’s model, and provide data to support that these activities. Overall the author provide a good body of work however, I have some suggestions that I believe would greatly improve this work.

Reviewer 2 ·

Basic reporting

- The tables is confused, please correct the number of tables and describe the parameters, not used abbreviations;

- Besides the effect on the α-amylase and α-glucosidase activity, what other enzymes or metabolic pathways could the compound be regulating? Stimulating the metabolism of carbohydrates in the liver, for example? Why? Please explain.

Experimental design

- The authors should include a figure of experimental design, to make clear the aim of the study;
-On which are carob concentrations based?
-Please, explain in the materials and methods why the used of nicotinamide to induce diabetes.
-Please add in all results (tables and figures) the experimental number.

Validity of the findings

No comments.

Additional comments

The study "Evaluation of the glycemic effect of Ceratonia siliqua pods (Carob) on nicotinamide-streptozotocin-induced diabetic rats model (#25216)" is designed well, the results are nicely summarized, and the findings are impressive and appropriate. This is a very nice article that needs only minor revision.

---

## Round 0.2 · accepted · Accept

I have received the re-reviews on your manuscript, " Evaluation of the glycemic effect of Ceratonia siliqua pods (Carob) on nicotinamide-streptozotocin-induced diabetic rats model".

Based on the advice received, I have decided that your manuscript could be accepted for publication in its present form.

# Reviewer 1 ·

Basic reporting

no comment

Experimental design

no comment

Validity of the findings

no comment

Additional comments

This manuscript describes the antioxidante and antihyperglycemic activity of the methanolic extract of Carob in nicotinamide-streptozotocin-induced diabetic rat’s model, and provide data to support that these activities.

Reviewer 2 ·

Basic reporting

No comments.

Experimental design

In the experimental design figure, I suggest used different colors in the parameters, for example: the pharmacological assays one color, toxicological and chemical another color.

Validity of the findings

No comments.